# Patient safety culture in resource-limited healthcare settings: A multicentre survey

**Gelana Fekadu**[1,2]*, **Rachel Muir**[1,3,4], **Georgia Tobiano**[5,6], **Aman Edao Bime**[7], **Michael J. Ireland**[8], **Andrea P. Marshall**[1,6]

1 School of Nursing and Midwifery, Griffith University, Gold Coast Campus, Southport, Queensland, Australia, 2 School of Nursing, College of Health and Medical Sciences, Haramaya University, Harar, Ethiopia, 3 Department of Emergency Medicine, Gold Coast Hospital and Health Service, Southport, Queensland, Australia, 4 Florence Nightingale Faculty of Nursing, Midwifery, and Palliative Care, Kings College, London, United Kingdom, 5 National Health and Medical Research Council, Centre of Research Excellence in Wiser Wound Care, Griffith University, Gold Coast Campus, Southport, Queensland, Australia, 6 Nursing and Midwifery Education and Research Unit, Gold Coast University Hospital and Health Service, Southport, Queensland, Australia, 7 Department of Anesthesiology, Critical Care, and Pain Medicine, College of Health and Medical Sciences, Haramaya University, Harar, Ethiopia, 8 School of Psychology and Wellbeing, University of Southern Queensland, Ipswich Campus, Ipswich, Queensland, Australia

⊕ These authors contributed equally to this work.
* fekadugelana4@gmail.com, gelana.mijena@griffithuni.edu.au

**Data availability statement:** All relevant data are within the paper and its Supporting information file (S2 dataset).

## Abstract

**Objective**: To assess healthcare professionals' perceptions of patient safety culture and to examine variations across clinical units in Eastern Ethiopian public hospitals.

**Methods**: A cross-sectional study was conducted using the Hospital Survey on Patient Safety Culture (HSoPSC 2.0) tool. Analysis of variance and ordinal logistic regression analyses were performed. Results were presented as mean differences and an adjusted odds ratio (AOR) with a 95% confidence interval (CI), and statistical significance was set at a p-value <0.05. Content analysis was performed for data provided through the open-ended response option.

**Results**: A total of 582 questionnaires were returned, yielding a response rate of 85%. Overall positive patient safety culture score was 47% (95% CI: 41–53%). Intensive care units (ICUs) scored significantly lower on patient safety culture dimensions compared to other clinical units. Factors contributing to the patient safety ratings included Midwives (AOR = 0.20, 95% CI: 0.06–0.71, p = 0.013), Organisational learning and continuous improvement (AOR = 1.35, 95% CI: 1.04–1.76, p = 0.025), Supervisor, manager, or clinical leader support for patient safety (AOR = 1.41, 95% CI: 1.06–1.89, p = 0.02), and Hospital management support for patient safety (AOR = 1.28, 95% CI: 1.00–1.63, p = 0.049). The challenges in ensuring patient safety included the absence of patient safety incident reporting systems, severe resource constraints, limited awareness regarding patient safety, ineffective communication, poor management support, and a blame-oriented organisational culture.

**Funding:** This study was funded by Griffith University International Postgraduate Research Scholarship program. The funding provided under the scholarship program for the first author. The funder had no role in study design, data collection and analysis, decision to publish, or preparation of the manuscript.

**Competing interests:** The authors have declared that no competing interests exist.

**Conclusions**: Significant improvement in patient safety culture in Ethiopian public hospitals, especially in the ICU, is critically needed to mitigate healthcare risks and ensure patient safety. Addressing these issues requires targeted patient safety training, strong leadership support, and adequate resource allocation. Further exploration of ICU-specific patient safety insights and validation of the HSoPSC 2.0 tool within the Ethiopian healthcare context should be undertaken to ensure cultural and contextual relevance.

## Introduction

Patient safety, defined as the prevention of harm associated with healthcare services, is a cornerstone of healthcare quality and has garnered increasing global attention [1,2]. Concerns about patient safety have increased owing to its direct impact on patient outcomes, significantly contributing to morbidity and mortality worldwide [3–5]. The Institute of Medicine and subsequent studies have identified unsafe care as a pervasive issue often exacerbated by suboptimal patient safety culture within healthcare institutions [6–8]. Patient safety culture is articulated as the shared values, beliefs, and norms within healthcare organisations that prioritise patient safety and shape the attitudes and behaviours of healthcare workers [9]. It reflects the commitment to identify and address patient safety risks, as well as the workplace habits that are expected, endorsed, and tolerated in relation to patient care [10]. A robust patient safety culture fosters shared responsibility and engagement in safety, which can be systematically measured and monitored [11].

Examining healthcare professionals' perceptions of patient safety culture is essential, as they are well positioned to identify unsafe practices and the contributing factors [12]. Such an assessment assists in pinpointing areas that require improvement and provides opportunities for benchmarking against best practices [13]. A well-developed patient safety culture is associated with favourable patient outcomes, higher patient satisfaction, shorter hospital stays, reduced healthcare costs, and a significant decrease in adverse events, morbidity, and mortality rates [14–16]. Furthermore, a positive and strong patient safety culture enhances staff job satisfaction and psychological safety, thereby enabling healthcare organisations to achieve the highest standards of safety and quality in patient care [17,18].

Patient safety culture encompasses various dimensions that impact safety practices, including teamwork, organisational learning and continuous improvement, effective feedback mechanisms, and open communication [19]. It promotes a blame-free environment for error reporting, fosters just culture in response to errors, ensures adequate staffing, maintains strong safety leadership, and facilitates seamless handoffs and information exchange [20]. These dimensions provide a framework for assessing and understanding patient safety culture, but contextual variations underscore the importance of tailored approaches [21]. The need for diligent evaluations further emphasises the importance of selecting appropriate measurement tools, as these influence the scope and relevance of the dimensions assessed. Although no single tool fully captures all aspects of patient safety culture [22].

The HSoPSC 1.0, developed by the Agency for Healthcare Research and Quality (AHRQ) in 2004, was widely used around the world [23]. The updated version, HSoPSC 2.0, released in 2019, introduced several key improvements, including refined wording, fewer negatively worded items, and a reduction in the number of items from 42 to 32 and dimensions from 12 to 10, resulting in a streamlined and user-friendly instrument. The updated version also demonstrates improved psychometric properties, places greater emphasis on staff input, and incorporates a just culture framework for assessing response to error. Furthermore, clear

guidance for result interpretations and benchmarking, along with its comprehensiveness and alignment with contemporary patient safety perspectives, make it the preferred tool [24,25]. HSoPSC 2.0 has been adopted in 62 countries [26] and has yielded acceptable validity and varying levels of reliability in various countries, including Brazil [27], China [28], Italy [29], India [30], Indonesia [31], Malaysia [32], Norway [33], South Korea [34], and Türkiye [35], although some scales showed weakness and require contextual modifications, as reported in studies from Australia [36] and Chile [37].

Despite increasing global attention, patient safety research remains limited in low- and middle-income countries (LMICs), where the burden of unsafe care is disproportionately high [2,4]. Systemic challenges such as inadequate infrastructure, skilled workforce shortages, weak governance, limited training opportunities, and disparities in patient safety awareness further undermine consistent safety standards and contribute to avoidable harm [38]. Ethiopia's healthcare system, in particular, faces significant challenges, including limited funding, underdeveloped infrastructure, persistent health inequities, pronounced regional disparities, and recurring sociopolitical instabilities, placing considerable pressure on delivering equitable, safe, and quality healthcare [39,40]. Although patient safety has recently gained recognition in Ethiopia, research in this area remains scarce. Recent studies [41,42] continue to rely on the outdated HSoPSC 1.0, despite the newer version being the current and recommended tool for assessing patient safety culture [24]. To the best of our knowledge, no prior study has evaluated patient safety culture using HSoPSC 2.0 in Ethiopian public hospitals. These indicated a significant gap in understanding the contemporary and comprehensive perspectives of healthcare professionals on patient safety culture in Ethiopian healthcare settings. Moreover, there is a lack of evidence on variations in patient safety culture across clinical units, despite its importance for developing targeted and unit-specific interventions [43]. In Ethiopia, as in many LMICs, the lack of context-specific research continues to impede the development of effective evidence-based strategies to improve patient safety. As such, research in these underexplored settings offers a valuable opportunity to generate locally relevant insights that can inform policy, practice, and future research. Addressing this gap is essential to foster safer healthcare environments, reduce preventable patient harm, and build resilient healthcare systems in resource-limited settings [44]. Therefore, the aim of this study was to assess healthcare professionals' perceptions of patient safety culture and examine potential differences across clinical units in Eastern Ethiopian public hospitals.

## Methods and materials

### Study design

A cross-sectional study was conducted. To ensure transparency, rigor, and reproducibility, the study adhered to the consensus-based checklist for reporting survey studies (CROSS) guidelines [45].

### Study settings and population

This study included five public hospitals located in Eastern Ethiopia, collectively serving a population of more than 8 million. Participating hospitals included two tertiary hospitals (A and B) and three secondary or general hospitals (C, D, and E). The hospital names were replaced by alphabetical identifiers to ensure anonymity. All five hospitals are situated approximately 520 km from the capital, Addis Ababa. Together, these hospitals employ over 1,400 healthcare professionals and collectively have 961 beds (range: 91 to 305 beds). In 2023, these

hospitals recorded 48,334 inpatient admissions and 784,586 visits to emergency and outpatient departments (EOPDs), underscoring their critical role in the regional healthcare system. The study population consisted of healthcare professionals (nurses, physicians, midwives, pharmacists, lab technologists, anesthesiologists, health officers, psychiatry nurses, and radiologists). To be eligible, participants were required to have a minimum of six months of work experience within the hospital and hold at least a bachelor's degree. Undergraduate students on clinical placement were not included.

## Sample size and sampling technique

The required sample size was determined to be 686 by using a single population proportion formula, considering 95% CI ($Z = 1.96$) and margin of error ($d = 0.447$), assuming a proportion of patient safety ratings as good (50.8%) from a previous study [46], and anticipating a response rate of 70%. To ensure representation, the sample size was proportionally allocated to each hospital based on the total number of healthcare professionals employed, using the formula: $Sample\ from\ hospital\ X = (\frac{Total\ number\ of\ healthcare\ professionals\ in\ Hospital\ X}{Total\ number\ of\ healthcare\ professionals\ in\ all\ Hospitals}) \times 686$. Details of the proportional allocation of participants are presented under supporting information S1 Table. Participating hospitals were purposively selected based on their operational capacity, diversity of services, willingness to participate, feasibility of data collection, and resource availability. To recruit participants, a consecutive sampling technique was employed, whereby all eligible healthcare professionals available during the data collection period were recruited until the target sample size was achieved. This approach was chosen for its practicality and suitability to the Ethiopian healthcare context, allowing for efficient recruitment without the need for a complete sampling frame (e.g., a full list of healthcare professionals in each clinical unit), unlike other probability sampling such as simple random sampling. Furthermore, consecutive sampling helps to minimise selection bias by avoiding subjective selection and including all accessible participants. It ensured that the sample reflected the routine flow of participants at the study sites and enhanced representativeness within limited financial and time availability [47].

## Data collection methods

Data were collected from July 28 to September 30, 2024, using a self-administered survey method. Given that English is the medium of instruction in Ethiopian higher education institutions and the working language in hospitals, structured questionnaires were provided in English. Healthcare professionals returned the completed questionnaires either directly to the research assistants or in secured boxes located in their clinical areas. The boxes were emptied daily. The survey was conducted anonymously, with no collection of participants' names or other personal identifiers. Upon return of the completed surveys, a unique identifier (ID) was assigned to each questionnaire before data entry was performed. Data collection was facilitated by two research assistants with prior experience.

## Survey instrument and data quality control

Sociodemographic characteristics of the participants were collected, including sex, age, educational status, professional role, work experience (in the hospital and unit), working unit, and working hours per week.

The HSoPSC 2.0 tool was utilised with written permission from AHRQ (CRM:00910086). The original HSoPSC 1.0 was updated in 2019 following users' feedback, resulting in the

development of the HSoPSC 2.0. The HSoPSC 2.0 encompasses 32 items across 10 composite dimensions (each comprising 2-4 items). Responses for each item are recorded on a 5-point Likert scale, measuring either agreement ('Strongly disagree' = 1 to 'Strongly agree' = 5) or frequency ('Never' = 1 to 'Always' = 5). Items are both positively and negatively worded. Additionally, the tool includes two single-item measures: patient safety ratings, assessed on a Likert scale ('Poor' to 'Excellent'), and the number of patient safety incidents reported in the past 12 months, categorised into five levels: 'None,' '1 to 2,' '3 to 5,' '6 to 10,' and '11 or more' [24]. The HSoPSC 2.0 has shown varying levels of validity and reliability across diverse contexts. In Australia [36] and Chile [37], refined models (26-item and 23-item, respectively) supported its construct validity and reliability. While the Brazilian version demonstrated solid construct validity, it showed weak internal consistency in dimensions like Handoffs and information exchange $\alpha$ = 0.50 and Staffing and work pace $\alpha$ = 0.41 [27]. Similarly, strong reliability was observed in Norway, except for the Teamwork dimension [33]. These findings highlight the tool's adaptability, though certain dimensions may require context-specific adjustments. Nonetheless, the HSoPSC 2.0 was not yet validated in the Ethiopian healthcare settings.

To ensure data quality, several approaches were implemented, beginning with a review of HSoPSC 2.0 before the commencement of data collection. The review team involved seven senior clinicians (four nurses and three physicians) with an average of 9.43 $\pm$ 3.16 years of work experience. The aim of this review was to verify the contextual relevance and clarify the instructions and wording of the items. Consistent with the tool developer's recommendations [24], minor modifications were made to the background questions to align with the local context. These included adjustments to staff positions (e.g., retaining only applicable roles and adding relevant ones) and units (e.g., changing "units" to "wards" and listing only those relevant to the local settings). Additionally, data on participant age, years of work experience, and weekly working hours were collected using open-ended questions or as continuous variables with responses specified in years, despite these being categorical in the original tool. The reviewer found the core questions of the tool to be clear and did not recommend any modifications.

Explicit instructions were provided to guide participants in completing and returning the survey. Research assistants received structured training prior to data collection. The training covered the purpose and objectives of the study, data collection procedures, key ethical considerations, and responding to participant questions without leading or influencing their responses. In addition, the training included a thorough review of the questionnaire and practical exercises to ensure consistency and standardisation during data collection. The first author G.F. (G.F. refers to Gelana Fekadu, first author of this study.) closely supervised the data collection process, addressing any recruitment or data collection challenges. Issues that could not be immediately resolved were escalated to the research team for further action. Data were collected using paper-based questionnaires that were checked for completeness and consistency before data entry. The data were entered in Epidata version 3.1 (Epidata Association, Odense, Denmark) by G.F. After entry, 15% of the data were randomly selected by using ID for rechecking against the paper surveys by R.M. (R.M. refers to Rachel Muir, co-author of this study.).

## Data analysis

All analyses were conducted using IBM SPSS Statistics for Windows version 30.0 (IBM Corp., Armonk, NY, USA). Missing data (2.2%) were addressed using mean substitution [48,49]. The data were approximately normally distributed, as skewness and kurtosis values for all variables

fell within the acceptable range of $\pm$ 1.50. Descriptive statistics, including frequency, percentage, mean, and standard deviation ($\pm$), were used to present the sociodemographic characteristics of the participants. The percentage of positive responses for each item and patient safety culture dimension was calculated according to the HSoPSC 2.0 user's guide [24]. For positively worded items, responses of 'Agree'= 4 and 'Strongly agree'= 5 were considered positive, whereas for negatively worded items, responses of 'Disagree' = 1 and 'Strongly disagree' = 2 were treated as positive and reverse coded. Composite-level scores were computed by summing the positive responses within each composite scale and dividing the total score by the number of items. Patient safety culture dimension scores were categorised as developed (>75% positive responses), poorly developed (50–74% positive responses), and underdeveloped (<50% positive responses) [24,50].

A series of one-way analysis of variance (ANOVA) was conducted to compare the mean scores of patient safety culture dimensions across clinical units. Levene's test assessed the homogeneity of variances, and since the assumption was not met, the Welch's test was applied [51]. To control for the familywise error rate in multiple comparisons, including pairwise comparisons, the Holm-Bonferroni correction was used [52]. Games-Howell post-hoc analyses were conducted, and the accompanying effect sizes, eta squared ($\eta^2$) were calculated to indicate the magnitude of the observed differences [53]. Ordinal logistic regression analyses were performed to identify factors predicting patient safety ratings. Assumptions such as multicollinearity among the independent variables were tested using the variance inflation factor (VIF) and the pairwise Pearson correlation matrix (r). A VIF >10 and r >0.80 were thresholds suggesting multicollinearity [54]. The assumptions were met with VIF<5 and r<0.66. Model fit and explanatory power were assessed using the Pearson Chi-square test ($\chi^2$) and multiple pseudo-$R^2$ measures were reported, including Cox and Snell ($R^2_{CS}$), Nagelkerke ($R^2_N$), and McFadden ($R^2_{MF}$) indices. Additionally, the proportional odds assumption was tested using a parallel line test, which confirmed the assumption was met (p>0.05) [55]. Content analysis and descriptive statistics were conducted for data provided through the open-ended response option [56]. The data used for analyses can be accessed from the supporting information file S2 dataset.

## Ethical considerations

The study adhered to the principles of the Declaration of Helsinki [57]. Ethical approval was obtained from the Institutional Health Research Ethics Review Committee (IHRERC) of the College of Health and Medical Sciences of Haramaya University, Ethiopia (IHRERC/181/2024), and the Human Research Ethics Committee of Griffith University, Australia (GU Ref No: 2024/453). A request letter for data collection permission was sent to all participating hospitals. Additionally, permission was obtained from each hospital medical director before the commencement of data collection.

Participants received an information sheet and consent form outlining the study's purpose, procedures, and duration of the survey; voluntary participation; participant anonymity; data confidentiality, storage, and security; the right to withdraw at any time without consequences; potential benefits and risks; and contact informations of the researcher team and office of ethics committees in case of any questions or concerns. Participants received the questionnaire after confirming their voluntary participation by returning the signed consent form while keeping the other copy for their records. For participants who preferred not to sign a written consent form, verbal consent was obtained after reading the full information sheet. This process was witnessed by research assistants and documented in a verbal consent log, including the anonymous code, date, and a confirmation of consent, without linking to

any personal identifiers. Participants could withdraw by simply not returning the completed questionnaire. The study did not include minors (individuals under 18 years of age).

## Results

### Sociodemographic characteristics of participants

Overall, 686 questionnaires were distributed, and 582 completed surveys were returned, representing a response rate of 85%. A total of 562 (96.6%) participants were engaged in direct patient care. The mean age of participants were 31.5 ± 6 years, ranging from 22 to 58 years, and over half, 324 (55.7%) were male. Most participants, 474 (81.4%) held a bachelor's degree. The mean work experiences in the current hospital and unit were 6.7 ± 5.1 and 3.7 ± 3.6 years, respectively. Most participants were nurses, 281 (48.3 %) followed by physicians, 153 (26.3%). Participants' working units included ICUs 128 (22.0%), medical wards 90 (15.46%), surgical wards 87 (14.9%), EOPDs 101 (17.4%), and Obstetrics and Gynaecology (Ob/Gyn) wards 78 (13.4%), as outlined in Table 1.

### Dimensions of patient safety culture

The overall positive patient safety culture score was 47% (95% CI: 41–53%). Among the ten dimensions, the highest positive response rate was observed for Teamwork (74%), followed by Organisational learning and continuous improvement (53%). In contrast, Reporting patient safety incidents received the lowest score (35.5%). The internal reliability, assessed using Cronbach's alpha ( $\alpha$ ), was above or equal to 0.6 for six dimensions, meeting the developer's threshold (Table 2).

### Patient safety rating and incident reporting

Nearly one-third, 187 (32%) of participants rated patient safety in their unit or work area as 'poor,' 91 (16%) as 'fair,' 181 (31%) as 'good', and 123 (21%) as 'very good.' Over the past 12 months, 326 (56%) participants had never reported patient safety incidents, whereas 145 (25%) reported one to two incidents (Fig 1).

### Patient safety culture dimensions across clinical units

One-way ANOVA showed significant differences in patient safety culture dimensions across clinical units (Welch's F range = 3.38–8.02, p <.003) except for Staffing and workplace. Effect sizes range from small $0.01 < \eta^2 < 0.06$ to medium, $0.06 < \eta^2 < 0.14$ as detailed in supporting information file S3 Table. However, no significant variations were observed across hospitals. Games-Howell post hoc test results revealed that the mean scores for each patient safety culture dimension were significantly lower in the ICUs compared to other clinical units, as presented in Table 3.

### Factors contributing to the patient safety rating

Bivariate ordinal logistic regression was conducted to examine the associations between the outcome variable, patient safety rating (categorised as poor, fair, good, and very good), and independent variables. Factors significantly associated with outcome variables at the bivariate level (p<0.05) were included in the multivariate ordinal logistic regression model. These encompassed sociodemographic characteristics (work experience in the current unit and professional role) and patient safety culture dimensions: Teamwork, Staffing and work pace,

**Table 1. Sociodemographic characteristics of the participants.**

| Sociodemographic Characteristics | Frequency | Percentage (%) |
|---|---|---|
| **Hospitals** | | |
| A | 250 | 43.0 |
| B | 84 | 14.4 |
| C | 67 | 11.5 |
| D | 112 | 19.2 |
| E | 69 | 11.9 |
| **Sex** | | |
| Male | 324 | 55.7 |
| Female | 258 | 44.3 |
| **Educational Status** | | |
| Bachelor's degree | 474 | 81.4 |
| Graduate degree | 108 | 18.6 |
| **Working Hours per Week** | | |
| <30 hours | 4 | 0.7 |
| 30–40 hours | 133 | 22.9 |
| >40 hours | 445 | 76.4 |
| **Work Experience in Hospital** | | |
| <1 year | 15 | 2.6 |
| 1–5 years | 295 | 50.7 |
| 6–10 years | 162 | 27.8 |
| ≥11 years | 110 | 18.9 |
| **Work Experience in Current Unit** | | |
| <1 year | 131 | 22.5 |
| 1–5 years | 329 | 56.5 |
| ≥6 years | 122 | 21.0 |
| **Professional Role** | | |
| Nurses | 281 | 48.3 |
| Physicians | 153 | 26.3 |
| Midwives | 59 | 10.1 |
| Pharmacists and lab technicians | 50 | 8.6 |
| Clinical team leaders and other staff* | 39 | 6.7 |
| **Providing Direct Patient Care** | | |
| Yes | 562 | 96.6 |
| No | 20 | 3.4 |
| **Working Units** | | |
| Medical ward | 90 | 15.5 |
| Surgical ward | 87 | 14.9 |
| Ob/Gyn ward | 78 | 13.4 |
| Paediatric ward | 43 | 7.4 |
| EOPD | 101 | 17.4 |
| ICU | 128 | 22.0 |
| PD | 55 | 9.5 |

*Note: * Other staff includes radiologists, anesthesiologists, health officers, psychiatry nurses, Ob/Gyn = Obstetrics and Gynaecology; EOPD = Emergency and Outpatient Department; ICU = Intensive Care Unit; PD = Pharmacy and Diagnostic Unit.*

Organisational learning and continuous improvement, Response to error, Supervisor, manager, or clinical leader support for patient safety, Communication about error, Communication openness, Reporting patient safety incidents, Hospital management support for patient safety, and Handoffs and information exchange.

The ordinal logistic regression model demonstrated a significantly better fit than the null model, $\chi^2(22) = 195.89$, $p < .001$, indicating that the predictors reliably distinguished between outcome categories. Goodness-of-fit statistics further supported the model's adequacy, with

**Table 2. Average percentage of positive response across composite dimensions and individual items.**

| Composite dimensions and items | Item description | Percentage of positive responses* | Cronbach's $\alpha$ |
|---|---|---|---|
| **Teamwork** | | 74 | 0.56 |
| C1 | In this unit, we work together as an effective team. | 85 | |
| C8 | During busy times, staff in this unit help each other. | 84 | |
| C9 (r) | There is a problem with disrespectful behaviour by staff working in this unit. | 53 | |
| **Staffing and work pace** | | 46 | 0.19 |
| C2 | In this unit, we have enough staff to handle the workload. | 45 | |
| C3 (r) | Staff in this unit work longer hours than is best for patient care. | 17 | |
| C5 (r) | This unit relies too much on temporary, float, or PRN staff. | 79 | |
| C11 (r) | 11. The work pace in this unit is so rushed that it negatively affects patient safety. | 45 | |
| **Organisational learning and continuous improvement** | | 53 | 0.68 |
| C4 | This unit regularly reviews work processes to determine if changes are needed to improve patient safety. | 61 | |
| C12 | In this unit, changes to improve patient safety are evaluated to see how well they worked. | 60 | |
| C14 (r) | This unit lets the same patient safety problems keep happening. | 38 | |
| **Response to error** | | 43 | 0.60 |
| C6 (r) | In this unit, staff feel like their mistakes are held against them. | 50 | |
| C7 (r) | When an event is reported in this unit, it feels like the person is being written up, not the problem. | 33 | |
| C10 | When staff make errors, this unit focuses on learning rather than blaming individuals. | 54 | |
| C13 (r) | In this unit, there is a lack of support for staff involved in patient safety errors. | 34 | |
| **Supervisor, manager, or clinical-leader support for patient safety** | | 48 | 0.52 |
| D1 | My supervisor, manager, or clinical leader seriously considers staff suggestions for improving patient safety. | 55 | |
| D2 (r) | My supervisor, manager, or clinical leader wants us to work faster during busy times, even if it means taking shortcuts. | 34 | |
| D3 | My supervisor, manager, or clinical leader takes action to address patient safety concerns that are brought to their attention. | 55 | |
| **Communication about error** | | 48 | 0.80 |
| E1 | We are informed about errors that happen in this unit. | 41 | |
| E2 | When errors happen in this unit, we discuss ways to prevent them from happening again. | 54 | |
| E3 | In this unit, we are informed about changes that are made based on event reports. | 48 | |
| **Communication openness** | | 42 | 0.70 |
| E4 | In this unit, staff speak up if they see something that may negatively affect patient care. | 47 | |
| E5 | When staff in this unit see someone with more authority doing something unsafe for patients, they speak up. | 40 | |
| E6 | When staff in this unit speak up, those with more authority are open to their patient safety concerns. | 41 | |
| E7 (r) | In this unit, staff are afraid to ask questions when something does not seem right. | 41 | |
| **Reporting patient safety incidents** | | 35.5 | 0.69 |
| G1 | When a mistake is caught and corrected before reaching the patient, how often is this reported? | 37 | |
| G2 | When a mistake reaches the patient and could have harmed the patient but did not, how often is this reported? | 34 | |
| **Hospital management support for patient safety** | | 42 | 0.64 |
| H1 | The actions of hospital management show that patient safety is a top priority. | 55 | |
| H2 | Hospital management provides adequate resources to improve patient safety. | 41 | |
| H3 (r) | Hospital management seems interested in patient safety only after an adverse event happens. | 31 | |
| **Handoffs and information exchange** | | 39 | 0.48 |
| H4 (r) | When transferring patients from one unit to another, important information is often left out. | 28 | |
| H5 (r) | During shift changes, important patient care information is often left out. | 30 | |
| H6 | During shift changes, there is adequate time to exchange all key patient care information. | 59 | |
| Overall | | 47 | 0.91 |

*Note: * = Mean percentage of positive responses calculated according to HSoPSC 2.0 user's-guide instructions; "r" = reverse-coded (negatively worded) items; PRN = pro re nata ("as needed").*

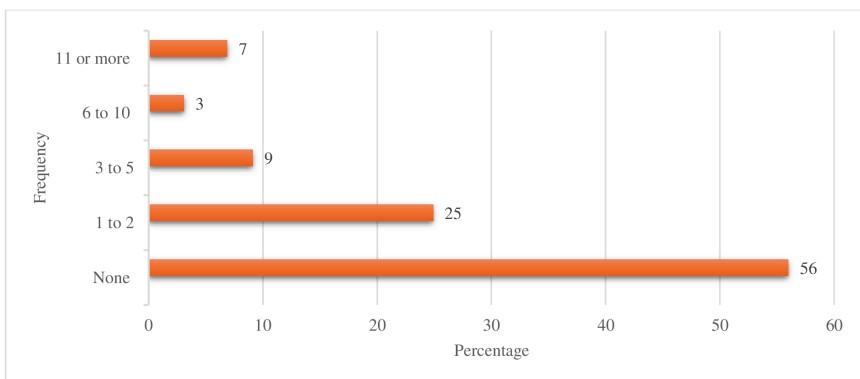

**Fig 1. Patient safety incidents reported in the past 12 months.**

a non-significant Pearson test ($p$ = .654), suggesting no substantial discrepancies between observed and predicted values. Pseudo-$R^2$ values indicated that the model explained approximately 29% of the variance in the outcome according to $R^2_{\mathrm{CS}}$, 30.7% according to $R^2_{\mathrm{N}}$, and 12.5% according to $R^2_{\mathrm{MF}}$. Overall, the model demonstrated a moderate level of explanatory power and a good fit to the data.

Based on the multivariate ordinal logistic regression analysis, midwives were significantly less likely to rate patient safety positively compared to clinical team leaders and other staff. The adjusted odds ratio (AOR) indicated a 79.2% lower likelihood of a positive rating among midwives (AOR = 0.20; 95% CI: 0.06–0.71; $p$ = 0.013). However, several factors were found to increase the likelihood of positive patient safety ratings. A one-unit increase in the mean score of the Organisational learning and continuous improvement dimension predicted a 35% greater likelihood of higher patient safety ratings (AOR = 1.35; 95% CI: 1.04–1.76; $p$ = 0.025). Similarly, a one-unit increase in the mean score of Supervisor, manager, or clinical leader support for patient safety predicted a 41% greater likelihood of higher ratings (AOR = 1.41; 95% CI: 1.06–1.89; $p$ = 0.02). A one-unit increase in the mean score for Hospital management support for patient safety predicted a 28% greater likelihood of a positive safety rating (AOR = 1.28; 95% CI: 1.00–1.63; $p$ = 0.049) (Table 4).

## Challenges in ensuring patient safety

The HSoPSC 2.0 included an open-ended comments section, allowing participants to share their perspectives on patient safety within their respective units and hospitals. Sixty participants (10%) provided feedback, with the most frequently mentioned concern being the lack of a patient safety incident reporting system by 17 (28%) participants. This was followed by severe resource constraints, including basic medical supplies and equipment, described by 13 (22%) participants, and poor management support and a blame-oriented organisational culture, noted by 12 (20%) participants. Moreover, 11 (18%) participants indicated a lack of training and poor awareness regarding patient safety, whereas 7 (12%) participants emphasised poor communication and information exchange.

## Discussion

Evaluation of healthcare professionals' perceptions on patient safety culture provides valuable insights on strengths and areas requiring improvement in patient safety. The aim of this

**Table 3. Comparison of patient safety culture dimensions across clinical units.**

| Dimensions | Clinical units | Mean difference [95% CI] | P-value |
|---|---|---|---|
| **Teamwork** | | | |
| | ICU vs Medical Ward | −0.370 [−0.679, −0.042] | 0.016 |
| | ICU vs EOPD | −0.420 [−0.711, −0.129] | < 0.001 |
| | Surgical Ward vs EOPD | −0.328 [−0.657, −0.003] | 0.046 |
| **Organisational learning and continuous improvement** | | | |
| | ICU vs Medical Ward | −0.818 [−1.250, −0.385] | < 0.001 |
| | ICU vs Ob/Gyn Ward | −0.533 [−0.966, −0.100] | 0.006 |
| | ICU vs PD | −0.669 [−1.118, −0.220] | < 0.001 |
| | ICU vs EOPD | −0.631 [−1.027, −0.237] | < 0.001 |
| | Surgical Ward vs Medical Ward | −0.628 [−1.106, −0.151] | 0.002 |
| **Response to error** | | | |
| | ICU vs EOPD | −0.243 [−0.457, −0.030] | 0.014 |
| **Supervisor, manager, or clinical leader support for patient safety** | | | |
| | ICU vs Medical Ward | −0.519 [−0.883, −0.154] | < 0.001 |
| | ICU vs PD | −0.519 [−0.924, −0.114] | 0.004 |
| | ICU vs EOPD | −0.569 [−0.910, −0.228] | < 0.001 |
| **Communication about error** | | | |
| | ICU vs Ob/Gyn Ward | −0.603 [−1.094, −0.113] | 0.006 |
| | ICU vs PD | −0.689 [−1.188, −0.189] | 0.001 |
| | ICU vs EOPD | −0.698 [−1.101, −0.295] | < 0.001 |
| **Communication openness** | | | |
| | ICU vs Medical Ward | −0.466 [−0.894, −0.039] | 0.022 |
| | ICU vs Ob/Gyn Ward | −0.477 [−0.912, −0.042] | 0.021 |
| | ICU vs PD | −0.613 [−1.061, −0.165] | 0.001 |
| | ICU vs EOPD | −0.653 [−1.029, −0.277] | < 0.001 |
| **Reporting patient safety incidents** | | | |
| | ICU vs EOPD | −0.426 [−0.871, −0.035] | 0.023 |
| **Hospital management support for patient safety** | | | |
| | ICU vs Medical Ward | −0.434 [−0.840, −0.029] | 0.027 |
| | ICU vs PD | −0.647 [−1.074, −0.220] | < 0.001 |
| | ICU vs EOPD | −0.430 [−0.782, −0.079] | 0.006 |
| **Handoffs and information exchange** | | | |
| | ICU vs Medical Ward | −0.516 [−0.847, −0.185] | < 0.001 |
| | ICU vs Ob/Gyn Ward | −0.473 [−0.799, −0.147] | < 0.001 |
| | ICU vs PD | −0.628 [−1.043, −0.214] | < 0.001 |
| | ICU vs EOPD | −0.471 [−0.763, −0.179] | < 0.001 |
| | Medical Ward vs Surgical Ward | 0.399 [0.484, 0.7505] | 0.015 |
| | Surgical Ward vs PD | −0.511 [−0.941, −0.082] | 0.009 |
| | Surgical Ward vs EOPD | −0.354 [−0.669, −0.039] | 0.017 |

*Note: CI = Confidence interval; EOPD = Emergency and outpatient department; ICU = Intensive care unit;*
*Ob/Gyn = Obstetrics and gynaecology; PD = Pharmacy and diagnostic unit; vs. = versus.*

study was to assess the healthcare professionals' perspectives on patient safety culture and to examine differences across clinical units using the HSoPSC 2.0 tool. The overall positive rating for patient safety was low at 47%. The dimension, Teamwork, received the highest score (74%) but remained below the 75% threshold to be considered developed. Patient safety event reporting was the lowest-rated dimension at 35.5%. Additionally, 56% of participants never reported patient safety incidents, and only 21% rated patient safety in their work unit as 'very good.' The mean scores for each patient safety culture dimension were compared across clinical units to identify areas for targeted interventions. All dimension scores were lower in

**Table 4. Multivariate ordinal-logistic regression showing factors predicting patient-safety rating.**

| Variables | Estimate | 95% CI | AOR | 95% CI | *P-value* |
|---|---|---|---|---|---|
| **Patient-safety-culture dimensions** | | | | | |
| Teamwork | 0.057 | [−0.213, 0.326] | 1.059 | [0.808, 1.385] | *0.680* |
| Staffing & work pace | −0.116 | [−0.431, 0.198] | 0.890 | [0.649, 1.219] | *0.468* |
| Organisational learning & continuous improvement | 0.301 | [0.037, 0.566] | 1.351 | [1.038, 1.761] | *0.025** |
| Response to error | −0.024 | [−0.393, 0.345] | 0.976 | [0.675, 1.412] | *0.900* |
| Supervisor/manager/clinical-leader support for patient safety | 0.345 | [0.055, 0.635] | 1.412 | [1.057, 1.887] | *0.020** |
| Communication about error | 0.093 | [−0.124, 0.309] | 1.098 | [0.883, 1.363] | *0.401* |
| Communication openness | 0.165 | [−0.089, 0.419] | 1.180 | [0.915, 1.520] | *0.202* |
| Reporting patient-safety incidents | 0.169 | [−0.016, 0.354] | 1.184 | [0.984, 1.426] | *0.073* |
| Hospital-management support for patient safety | 0.244 | [0.001, 0.488] | 1.276 | [1.001, 1.629] | *0.049** |
| Handoffs & information exchange | 0.121 | [−0.137, 0.380] | 1.128 | [0.872, 1.462] | *0.358* |
| **Clinical units** | | | | | |
| Medical ward | −0.578 | [−1.182, 0.026] | 0.561 | [0.307, 1.027] | *0.061* |
| Surgical ward | −0.267 | [−0.874, 0.341] | 0.766 | [0.417, 1.407] | *0.389* |
| Ob/Gyn ward | 0.412 | [−0.616, 1.441] | 1.510 | [0.540, 4.225] | *0.432* |
| Paediatric ward | −0.182 | [−0.904, 0.540] | 0.834 | [0.405, 1.717] | *0.622* |
| ICU | −0.358 | [−0.928, 0.213] | 0.698 | [0.395, 1.237] | *0.219* |
| PD | 0.155 | [−1.037, 1.347] | 1.168 | [0.355, 3.848] | *0.799* |
| EOPD | Ref. | — | — | — | — |
| **Professional roles** | | | | | |
| Nurses | −0.477 | [−1.180, 0.226] | 0.623 | [0.307, 1.254] | *0.183* |
| Physicians | −0.676 | [−1.433, 0.081] | 0.509 | [0.239, 1.084] | *0.080* |
| Midwives | −1.569 | [−2.801, −0.337] | 0.208 | [0.060, 0.713] | *0.013** |
| Pharmacy & lab technicians | −0.725 | [−1.965, 0.516] | 0.483 | [0.140, 1.675] | *0.252* |
| Clinical team leaders & other staff | Ref. | — | — | — | — |
| **Work experience in current unit** | | | | | |
| <1 year | −0.414 | [−1.169, 0.342] | 0.660 | [0.312, 1.409] | *0.283* |
| 1–5 years | −0.726 | [−1.552, 0.101] | 0.483 | [0.212, 1.106] | *0.085* |
| ≥ 6 years | Ref. | — | — | — | — |

*Note: AOR = adjusted odds ratio; CI = confidence interval; EOPD = emergency and outpatient department; ICU = intensive care unit; Ob/Gyn = obstetrics and gynecology; PD = pharmacy and diagnostic unit; Ref. = reference category; * significant at p<0.05.*

ICUs compared to other clinical units. Furthermore, Organisational learning and continuous improvement, Supervisor, manager, or clinical leader support for patient safety, and Hospital management support for patient safety were predicted a higher likelihood of positive patient safety ratings. However, midwives were less likely to rate patient safety positively compared to clinical team leaders and other staff. Participants highlighted challenges in ensuring patient safety, including a lack of patient safety incident reporting systems, severe resource constraints, poor management support, a blame-oriented organisational culture, ineffective communication, and limited awareness regarding patient safety.

In this study, an overall positive patient safety culture score was 47% (95% CI: 41-53%), indicating an underdeveloped patient safety culture in Eastern Ethiopian public hospitals. This finding is consistent with local studies conducted between 2016 and 2024 across various regions of Ethiopia, which reported patient safety ratings ranging from 37% to 50.8% [41,46, 50,58–64]. These results indicate a persistently underdeveloped and stagnant level of patient safety in Ethiopian healthcare facilities. Moreover, these studies used the HSoPSC 1.0 tool,

despite the recommendations for the updated version, HSoPSC 2.0. Furthermore, benchmarking of the current study findings with the AHRQ user database of 2024 from USA hospitals [65], organisation for economic co-operation and development (OECD) countries, Saudi Arabia, and LMICs such as Brazil, China, India, and Malaysia reveals both shared strengths and persistent gaps as assessed by using the HSoPSC 2.0 (supporting information file, S4 Fig). In the USA and Brazil, the highest overall patient safety rating (71%) was reported [27,65] followed by China (68%) [28]. An overall poorly developed patient safety culture was reported across all countries, with scores ranging from 50% to 71%. In contrast, the current study, with a score of 47%, is classified as underdeveloped, highlighting a significant need for improvement at both the national and global levels. In this study, Teamwork was the highest positively rated dimension at 74%, followed by Organisational learning and continuous improvement at 53%, yet poorly developed. In contrast, upper-middle and HICs such as the USA [65], OECD [66], and China [28] showed stronger performance (>75%) in Teamwork, Organisational learning and continuous improvement, Supervisor, manager, or clinical leader support for patient safety, and Communication about error, indicating developed safety culture aspects. However, all countries, including HICs, reported low scores in Staffing and work pace and Response to error. This highlights that even well-resourced healthcare systems face challenges in workforce sustainability and error management, underscoring the universal need for targeted policies and system-level interventions to support staffing, reduce workload pressure, and promote a just culture [67]. The differences between HICs and LMICs highlight the significant disparities in resources and commitment to patient safety [68]. For instance, in Ethiopia, the chronically underdeveloped patient safety culture reflects a historical lack of prioritisation, as healthcare systems have traditionally focused on expanding service coverage rather than ensuring safety and quality [69]. While recent initiatives, such as the introduction of national patient safety guidelines, are promising, substantial efforts are still needed to fully integrate these into the healthcare system [70]. Recurrent socio-political instability, limited funding, weak governance, fragile and inefficient health insurance schemes, and entrenched low socio-economic conditions continue to pose significant challenges to healthcare quality and safety [71–73]. Strengthening the safety culture in Ethiopian hospitals and similar resource-limited settings requires targeted, cost-effective strategies that emphasise systemic reform and stakeholder engagement. Central to this effort is strong leadership committed to embedding patient safety as a core organisational priority. Furthermore, regular monitoring, evaluation, and benchmarking against national and international standards can further track progress and ensure sustainability [11].

In this study, lower positive scores in areas such as Handoffs and information exchange (39%), Reporting patient safety events (35.5%), Communication openness (42%), and Response to error (43%) indicated serious deficiencies in organisational transparency and psychological safety. In comparison, data from the AHRQ user database in the USA revealed developed status in Communication about error, Communication openness, and Reporting patient safety events [65]. Effective communication is a fundamental element of healthcare systems and plays a critical role in preventing medical errors and ensuring patient safety [74]. Therefore, it is essential to nurture psychological safety through open and respectful communication among frontline clinicians and managers utilising standard communication tools, such as SBAR (situation, background, assessment, and recommendation) [75]. Furthermore, providing staff training through evidence-based frameworks such as TeamSTEPPS (Team Strategies and Tools to Enhance Performance and Patient Safety) is essential [76]. Such frameworks can be used for targeted interventions or integrated into continuous professional development (CPD) programmes through e-learning approaches or using local expertise and on-the-job training sessions to ensure affordable and sustainable improvements. Consequently, in

teaching hospitals, where diverse professionals, including health and medical science students in clinical placements, internships, or residency programs, are involved in patient care, prioritising collaboration and effective communication is vital for promoting patient safety [77]. Actively involving frontline clinicians in the development of context-sensitive patient safety solutions strengthens the sense of ownership and adherence [78]. In addition, engaging patients and families as partners is crucial for building an inclusive and sustainable patient safety culture [8]. International collaboration and the adoption of best practices from high-performing countries can help to develop tailored solutions to foster patient safety culture in resource-limited healthcare settings [2].

This study revealed that only one-fourth of the participants reported one to two patient safety incidents in the past 12 months, while over half (56%) reported none. The patient safety event reporting dimension received the lowest positive score at 35.5%. These findings align with earlier local studies reporting rates between 25.4% and 31.9% [79,80], yet fall below the 53.2% reported in Uganda [81]. The persistently low incident reporting rates in Ethiopian public hospitals may stem from the absence of formal reporting systems, as well as a blame-and-punishment-oriented organisational culture. Congruently, underdeveloped communication-related dimensions reveal systemic flaws in safety incident reporting and responses. Similar patterns were observed in many LMICs where reporting systems were either absent or poorly implemented [82,83]. In contrast, in HICs, safety incident reporting rates often exceed 50%, through well-established systems and strong safety-oriented leadership, although underreporting persists in this contexts [84]. Fear, blaming culture, and punitive response continue to be major barriers to incident reporting globally [85], compounded by limited leadership commitment, inadequate feedback loops, inefficient systems, and weak teamwork [86,87]. These cultural and structural deficiencies erode trust and hinder the development of psychological safety, essential for open communication and learning from incidents [88]. Addressing these challenges requires establishing anonymous, cost-effective, and user-friendly reporting systems such as digital platforms (e.g., mobile applications) [89,90]. These efforts should be supported by leadership-driven initiatives to promote a just culture through the adoption of evidence-based models, such as the High Reliability Organisation (HRO) model [91], supported by regular staff training, visible feedback, integration of reporting into routine clinical practice, and recognition of reporting efforts, such as providing certificates or celebrating departments with better performance. In addition, a supportive policy framework to tackle fear-related challenges is essential to sustain reporting practices [92]. Such integrated approaches foster a culture of continuous learning and improvement in patient safety [87,93,95,95].

Our study results indicated that the mean scores for patient safety culture dimensions were lower in ICUs compared with other clinical units, which is consistent with the results of previous studies from Iran and Tunisia [96,97]. ICUs are complex settings in which critically ill patients require advanced monitoring and multidisciplinary care [98]. In Ethiopia, challenges in ensuring patient safety in ICUs are exacerbated by poor management support, inadequate infrastructure and supplies, high workloads, insufficient staffing, suboptimal capacity building, and continuous training opportunities [99]. Therefore, to enhance patient safety culture, it is essential to prioritise strong leadership support and designate safety champions, foster a blame-free and just culture, and simplify error reporting processes [86]. Building staff capacity through low-cost, ICU-specific trainings, such as brief, case-based discussions during bedside rounds, and shift handovers are vital. Standardising communication using SBAR, monitoring safety culture progress, and recognising staff contributions *(e.g., "Safety star of the month")* can further enhance engagement. Improving access to basic medical supplies and

equipment, such as monitors and essential medicines, along with enhancing working conditions by promoting low-cost hand hygiene stations, ensuring adequate personal protective equipment, enforcing infection prevention protocols, and managing traffic flow within the unit, is critical to strengthening patient safety in resource-limited settings [97]. Furthermore, the cohesive integration of people, processes, technology, and data is essential in ensuring safe care in ICUs in low-resource settings. These include investing in people by establishing multi-professional ICU teams, improving staff-to-patient ratios, and reducing workloads to minimise burnout. Strengthening the process through standardised workflows and evidence-based decision-making aids enhances the consistency and reliability of care [100]. Leveraging appropriate technology, suited to the local context, supports clinical decisions and operational efficiency. Additionally, the effective use of data for monitoring outcomes, identifying gaps, and guiding continuous improvement is vital for maintaining high standards of care [101]. Organisations should adapt global best practices to local realities to enhance safety outcomes and foster sustainable improvements in ICU performance [102].

In this study we identified a significant disparity in patient safety ratings among healthcare professionals, with Midwives being 79.2% less likely to rate patient safety positively than clinical team leaders and other staff. This finding highlights a critical gap in perceptions among professional groups, which may be linked to higher workloads with limited support and resources in maternity care settings, which increase burnout and a higher risk of safety incidents [103]. Similarly, a study conducted in Sweden found that midwives reported lower scores in patient safety compared to other professionals [104]. While an international study from Austria, Germany, and Switzerland found that midwives often experience discrepancy between the perceived importance of patient safety culture and its implementation in daily practice. This was attributed to the absence of supportive and organised incident management systems, challenges in interprofessional dynamics inhibiting the decision-making autonomy, insufficient education and training, and poor communications [105]. Additionally, shortages of essential medical equipment and supplies in maternity units create an environment where maintaining high patient safety standards is challenging, especially in resource-limited healthcare settings [106]. These findings underscore the need for targeted interventions guided by evidence-based frameworks, such as the Donabedian model, to identify and address structural, process, and outcome-related challenges in ensuring patient safety within the healthcare system [107]. In addition, key strategies such as simulation-based trainings, regular patient safety workshops, and e-learnings are vital. Promoting interprofessional collaboration through team huddles and shared case reviews, encouraging open communication via regular debriefings, and adopting a blame-free approach through anonymous incident reporting systems further support a culture of safety. Adequate funding for education, training, and staffing is also essential. Together, these measures can improve midwives' perceptions of patient safety, foster teamwork across professional groups, and enhance the quality of care [108,109].

This study demonstrated the key factors that predict positive patient safety ratings, including, Organisational learning and continuous improvement; Supervisor, manager, or clinical leader support for patient safety; and Hospital management support for patient safety, which were associated with 35%, 41%, and 28% likelihoods of positive patient safety culture ratings, respectively. These findings align with previous studies showing that strong leadership fosters transparency, builds staff trust, and improves patient safety outcome [110]. Continuous learning and improvement enhance safety [111], by actively involving management and frontline workers in patient safety initiatives, which in turn boosts staff morale, promotes adherence to safety practices, and fosters stewardship [112]. Hospital leaders must visibly support patient safety by allocating sufficient resources for staff training and recognising adherence to safety

protocols. In addition, they should shift focus from numerical metrics to quality- and safety-based performance monitoring [113]. For instance, in Ethiopian healthcare facilities, where evaluations often prioritise quantitative outputs *(e.g., the number of patients who received care and were discharged)*, greater emphasis should be placed on the number of patients who received safe and quality care before being discharged [114]. Overall, achieving lasting change requires a paradigm shift in how patient safety is perceived and implemented. This includes integrating patient safety into medical and health science curricula, as the study found limited awareness of these principles. Adopting a 'patient safety first' approach is also essential to ensure high-quality and safe care, contributing to global efforts to eliminate preventable patient harm by 2030 [2,115,116].

## Strengths and limitations

This study has several notable strengths. It employed the HSoPSC 2.0 tool, the updated version recommended by the AHRQ, which was designed for precise assessment of patient safety culture in healthcare settings. The use of this tool enables benchmarking against international studies and offers broader and up-to-date insights into patient safety culture. The inclusion of 582 healthcare professionals from diverse roles strengthened the study's representativeness, enriched the perspectives captured, and provided adequate statistical power to detect meaningful trends and associations. Moreover, the participation of five hospitals with varying capacities, including both secondary and tertiary institutions, enhanced the comprehensiveness of the findings, identifying key patterns and areas of concern across different facility types. The study's rigorous methodology also offers a replicable framework for similar research in other regions or for scaling up to a national-level survey. However, the study was limited to public hospitals in eastern Ethiopia, which may constrain the generalisability of the findings to the broader national context, especially to private or rural healthcare facilities.

A key methodological limitation concerns the low internal consistency observed in several HSoPSC 2.0 subscales. Cronbach's alpha coefficients ranged from 0.19 (Staffing and work pace) to 0.80 (Communication about error), with eight subscales falling below the commonly accepted threshold of 0.70 and four falling below the HSoPSC 2.0 developers' minimum criterion of 0.60. These low alpha values suggest potential measurement limitations, which may influence the reliability of the composite scores and should be considered when interpreting the findings. In particular, the Staffing and work pace ($\alpha = .19$) and Handoffs and information exchange ($\alpha = .48$) subscales showed notably poor internal consistency. These raise concern that those non-significant findings involving these dimensions, especially in the ANOVA and ordinal logistic regression analyses, may reflect insufficient reliability rather than a true absence of effect. Measurement error of this magnitude attenuates correlations and group differences, increasing the likelihood of Type-II errors and underestimation of effect sizes. Therefore, the null results for these subscales should be interpreted with caution, as important relationships may have been obscured by poor scale reliability. Future research should prioritise the refinement or replacement of low-performing subscales to enhance measurement precision and statistical power.

The questionnaires were administered in English. Although English is the medium of instruction in Ethiopian higher education institutions and the primary language of communication in hospitals, some staff members may find completing a questionnaire in English to some extent challenging, potentially impacting their understanding of certain items. Nevertheless, a previous study in Ethiopia that assessed patient safety culture using a local language reported a similar overall average positive score [64], suggesting that the language of the questionnaire may not have significantly influenced the results. This study focused solely

on healthcare professionals while excluding supportive and administrative staff (e.g., finance, janitors, and others) and professionals with an educational status below a bachelor's degree. This limitation may affect the comprehensiveness of the findings and warrants caution in interpretation, as patient safety relies on the contributions of all healthcare workers.

## Implications for policy, practice, and research

The findings of this study provided several practical recommendations for policy, practice, and future research to improve patient safety culture in resource-limited healthcare settings. These require strong political willingness, proactive leadership, and comprehensive policy frameworks [106]. Policies should prioritise a systematic integration of patient safety culture concepts from top to grassroots level in healthcare systems and active engagement of stakeholders, including healthcare managers, frontline healthcare workers, patients, and families [117]. Additionally, facilitating adequate resource allocation and efficient utilisation of funds through prioritising high-impact expenditure and strengthening healthcare financing mechanisms, such as health insurance schemes, are critical in addressing systemic challenges such as poor infrastructure and high workloads, particularly in ICUs [100,118]. These measures are essential in reducing disparities in patient safety across clinical units and enhancing the overall quality of care and clinical outcomes [119]. Additionally, ongoing monitoring and evaluations of indicators and outcomes are essential to inform further policy refinement and planning for sustainability [120].

Institutionalising continuous learning and improvement through the development of evidence-based guidelines for patient safety incident reporting systems is critical. These initiatives can be scaled by adapting global patient safety guides *(e.g., WHO's patient safety incident reporting systems guidelines)* to local contexts and integrating them into routine workflows [121]. Sustainability can be ensured through the development of policy frameworks advocating for a non-punitive reporting culture and linking reporting to routine clinical practices and quality improvement initiatives. Healthcare institutions should adopt user-friendly, simplified reporting systems that account for resource limitations and do not rely on expensive technologies or expertise, such as mobile applications or paper-based reporting systems [90]. Encouraging staff to report incidents, conducting root cause analyses, and providing timely, constructive feedback with balanced responses can further improve reporting practices [86,87]. Promoting interprofessional collaboration through team-based training programs using evidence-based frameworks, such as TeamSTEPPS, and using communication tools, such as SBAR, can improve teamwork and communication across clinical units with a special focus on ICUs [122]. Encouraging patient and family involvement in patient safety initiatives and recognising staff for adhering to safety protocols can create a harmonious working atmosphere and sustainable safety improvements [123]. Furthermore, integrating patient safety into medical and health science education curricula lays the groundwork for long-term cultural changes [124]. This can be achieved through collaboration with academic institutions and regulatory bodies to standardise patient safety concepts across disciplines. To maintain momentum, curricula should be periodically reviewed and aligned with current evidence, while graduates are supported by CPD and safety leadership opportunities [125].

Future research should prioritise longitudinal studies to track changes in patient safety culture over time and their impact on patient outcomes across diverse healthcare workers, with particular attention to ICUs. In-depth qualitative research exploring healthcare professionals' perceptions and experiences of patient safety and incident reporting in ICUs is essential to understand factors contributing to disparities in safety ratings and reporting behaviours. Additionally, research on the cultural adaptation and psychometric validation of the HSoPSC

2.0 is vital to ensure its contextual relevance. Interventional studies should also evaluate the effectiveness of targeted strategies to improve patient safety culture in resource-limited settings. Further, follow-up studies are needed to assess the scalability and sustainability of implemented strategies to ensure their effectiveness. Implementing these recommendations could help bridge the patient safety gap between low- and high-resource settings, supporting the WHO's goal of eliminating avoidable patient harm by 2030 and ensuring safe care for all [2].

## Conclusion

This study offers critical insights into the deeply inadequate state of patient safety culture in Ethiopian public hospitals, particularly severe deficiencies in ICUs. Inadequate patient safety incident reporting practices, likely stemming from the absence of structured and formal reporting systems, a blame- and punishment-oriented organisational culture, and limited management support, were identified. In contrast, key drivers of higher patient safety ratings included organisational learning and continuous improvement, effective clinical leadership, and strong managerial support. The study also revealed significant challenges in ensuring patient safety, including severe resource constraints, poor communication, and limited awareness of patient safety concepts.

Improving patient safety culture, especially within ICUs, requires a comprehensive and context-sensitive approach. This includes political willingness, strong commitment from policymakers and healthcare managers to design and implement accessible and cost-effective incident reporting systems, proactive safety promotion approaches such as conducting regular safety audits, and multidisciplinary training through affordable methods such as online platforms and using local expertise. Fostering a blame-free and non-punitive culture, promoting the psychological safety of staff, and recognising and rewarding safety compliance can further reinforce positive safety behaviours. Adapting insights from high-performing health systems and integrating them with local and context-sensitive evidence can enhance patient safety culture and improve healthcare resilience in resource-limited settings.

## Supporting information

**S1 Table. Proportional allocation of participants in each hospital, Eastern Ethiopia.**
(DOCX)

**S2 dataset. Dataset used for analyses.**
(SAV)

**S3 Table. The mean score of patient safety culture dimensions across clinical units.**
(DCOX)

**S4 Fig. Graph showing patient safety culture dimensions across various countries and regions.**
(TIF)

## Author contributions

**Conceptualization:** Gelana Fekadu, Rachel Muir, Georgia Tobiano, Andrea P. Marshall.

**Data curation:** Gelana Fekadu, Rachel Muir, Georgia Tobiano, Michael J. Ireland, Andrea P. Marshall.

**Formal analysis:** Gelana Fekadu, Michael J. Ireland, Andrea P. Marshall.

**Funding acquisition:** Gelana Fekadu, Andrea P. Marshall.

**Investigation:** Gelana Fekadu, Rachel Muir, Aman Edao Bime, Michael J. Ireland, Andrea P. Marshall.

**Methodology:** Gelana Fekadu, Rachel Muir, Georgia Tobiano, Michael J. Ireland, Andrea P. Marshall.

**Project administration:** Gelana Fekadu, Rachel Muir, Georgia Tobiano, Andrea P. Marshall.

**Resources:** Gelana Fekadu, Michael J. Ireland, Andrea P. Marshall.

**Software:** Gelana Fekadu, Rachel Muir.

**Supervision:** Rachel Muir, Georgia Tobiano, Aman Edao Bime, Michael J. Ireland, Andrea P. Marshall.

**Validation:** Gelana Fekadu, Rachel Muir, Aman Edao Bime, Michael J. Ireland, Andrea P. Marshall.

**Visualization:** Gelana Fekadu, Rachel Muir, Georgia Tobiano, Aman Edao Bime, Michael J. Ireland, Andrea P. Marshall.

**Writing – original draft:** Gelana Fekadu.

**Writing – review & editing:** Gelana Fekadu, Rachel Muir, Aman Edao Bime, Michael J. Ireland, Andrea P. Marshall.

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
