## [Decision Letter · Decision Letter 0]

21 Apr 2025

PONE-D-25-07741Patient safety culture in resource-limited settings: insights from healthcare professionals through a multi-centre survey.PLOS ONE

Dear Dr. Fekadu,

Thank you for submitting your manuscript to PLOS ONE. After careful consideration, we feel that it has merit but does not fully meet PLOS ONE’s publication criteria as it currently stands. Therefore, we invite you to submit a revised version of the manuscript that addresses the points raised during the review process.

**ACADEMIC EDITOR: **Please address the reviewers' comments, especially those related to benchmarking, methodological robustness and transparency and thematic integration. 

We look forward to receiving your revised manuscript.

Kind regards,

Hossam Elamir, MSc

Academic Editor

PLOS ONE

Journal Requirements:

3. Thank you for stating the following financial disclosure: [This study was funded by Griffith University International Postgraduate Research

Scholarship program. The funding provided under the scholarship program for the first author. No funding was received for the Article Processing Charge (APC)]. 

Reviewers' comments:

Reviewer's Responses to Questions

**Comments to the Author**

1. Is the manuscript technically sound, and do the data support the conclusions?

Reviewer #1: Yes

Reviewer #2: Yes

Reviewer #3: Partly

Reviewer #4: Yes

Reviewer #5: Yes

Reviewer #6: Partly

2. Has the statistical analysis been performed appropriately and rigorously? 

Reviewer #1: Yes

Reviewer #2: Yes

Reviewer #3: Yes

Reviewer #4: Yes

Reviewer #5: Yes

Reviewer #6: Yes

3. Have the authors made all data underlying the findings in their manuscript fully available?

Reviewer #1: No

Reviewer #2: No

Reviewer #3: Yes

Reviewer #4: Yes

Reviewer #5: Yes

Reviewer #6: No

4. Is the manuscript presented in an intelligible fashion and written in standard English?

Reviewer #1: Yes

Reviewer #2: Yes

Reviewer #3: No

Reviewer #4: Yes

Reviewer #5: Yes

Reviewer #6: Yes

5. Review Comments to the Author

Reviewer #1: I would like to thank you for submitting this insightful paper on patient safety culture. This is an incredibly important topic and your research provides valuable perspectives that can help advance the understanding and improvement of safety protocols in healthcare settings.

The paper is well-written and highlights several crucial aspects of patient safety. I do have a few minor comments, specifically regarding the sample size and some potential gaps in the research that could be explored further.

1. Sample size and sampling technique

Would you please clarify how the sample size was proportionally allocated to each hospital based on the total number of staff and how much was given to each hospital.

2. Conclusion

It is worth stating the gaps this study has covered and any other gaps that should be investigated in the future.

Reviewer #2: The paper provides valuable insights into patient safety culture in resource-limited settings, particularly in Ethiopian public hospitals. The study is well-designed, with a correct methodology responding to the study objectives. However there are some limitations such as:

• The low Internal Consistency of Subscales: Several subscales in the HSoPSC 2.0 showed poor internal consistency (Cronbach’s α < 0.70), which may have affected the reliability of the results. This limitation is acknowledged by the authors but could impact the validity of the findings.

• Language Barrier : Although English is the medium of instruction in Ethiopian hospitals, some participants may have faced challenges in understanding the questionnaire, potentially affecting their responses.

Reviewer #3: Thank you to look over my insight full expertise on this research work entitled “Patient safety culture in resource-limited settings: insights from healthcare professionals through a multi-center survey.” and I got this article is very interesting and thought full and identifying deferent Hedin factors that identified Patient safety culture in resource-limited settings.

The research work is very interesting and thought full for scholars and also leading factor for other researchers and also hospital administrators farther more for ministry of health

1. How do you measure Cronba ch’s α and what is the cut point of Cronba ch’s α?

2. What is your base line that you said that the patient safety is present or absent in the hospital? with reference

3. Why you were select the linear model for your research? Clearly enplane

4. Inline285 your finding “Compared with supervisors and other clinical staff, midwives were significantly less likely to rate patient safety positively,” why this is happened? Would you explain the tool you used (if you can tool).

5. Do you think that “lack of a patient safety incident reporting system,” is the main challenge is this how do you measure?

6. I do have question on your exclusion criteria do you think that the absent of study participant during study period exclusion criteria?

#. Do you think Is it possible two ethical clearance for one research work?

General comment for researchers I have read your finding and which is very good work but I recommended your study population was patients because the key stakeholder due to understand or answer your research title the actors were better to take patients

Reviewer #4: Dear sir,

The study examined ‘Patient safety culture in resource-limited settings: insights from healthcare professionals through a multi-centre survey’. This manuscript is well-written and sound, offering a significant understanding of patient safety culture in Ethiopian public hospitals. The study effectively identifies gaps in patient safety culture and provides actionable recommendations for improving it. However, several areas could be enhanced. Overall, with minor improvements, this manuscript has the potential to contribute significantly to the field of patient safety, particularly in resource-limited settings.

Introduction

The transition from HSoPSC 1.0 to 2.0 is mentioned, but the discussion on how HSoPSC 2.0 improves explicitly upon the previous version could be expanded. More concrete examples of its improvements, such as added dimensions or clarified questions, would enhance the reader’s understanding.

Again, while the introduction mentions Ethiopia’s under-researched state, a deeper exploration of the Ethiopian healthcare system’s specific challenges could enrich the introduction. For instance, referencing infrastructure deficits, regional disparities, and the socio-political climate in Ethiopia might provide more context for the gap in research.

Methods and Materials

1. The use of “consecutive sampling” is mentioned, but the rationale behind this choice could be better explained. For example, it could clarify why this technique minimizes selection bias and is suited for the context of Ethiopian public hospitals.

2. Was the tool pre-tested? While the HSoPSC 2.0 tool was adapted to local contexts, it would be helpful to discuss the specific modifications made to the tool (e.g., changes to the wording of questions or scale adaptation) to ensure its appropriateness for Ethiopian healthcare settings.

Results

1. The manuscript mentions “small to medium effect sizes (η²)” but does not provide specific values, which would help the reader interpret the magnitude of differences observed.

2. The manuscript should emphasize the practical significance of findings and statistical significance. For example, it could describe the implications of midwives’ lower ratings of patient safety culture beyond the statistical significance.

3. Why content analysis? Which questions demanded open-ended responses?

4. Which bivariate analysis was done? Correlation? Any results for that?

Discussion

1. While the study draws comparisons with other settings, a more profound synthesis of the results with existing theoretical frameworks on patient safety culture would strengthen the discussion. Connecting the results to safety culture models could add depth to the interpretation.

2. Some limitations, such as the poor internal consistency of specific subscales, are briefly mentioned. These should be more deeply integrated into the discussion, particularly in how they affect the interpretation of the findings and the robustness of the conclusions.

Strengths and Limitations

The internal consistency issues with several key subscales (e.g., Staffing and Work Pace) should be addressed more critically. A more precise explanation of how these limitations might have impacted the findings and their practical implications is needed.

Conclusion and Implications

While the conclusion outlines essential policy and practice recommendations, it could benefit from more emphasis on the long-term sustainability of these changes. For example, how might these recommendations be scaled or maintained over time, especially in resource-limited settings?

The call for future research could be more specific. For instance, it would be beneficial to suggest follow-up studies focusing on the long-term impact of the interventions recommended in this study.

Reviewer #5: Thank you for your important study on patient safety culture in Ethiopian hospitals.

It is a lovely, well-written study, and I wish to be published.

Your findings highlight critical gaps, especially in ICUs, and provide a strong foundation for improvement.

To build on this work, I recommend developing a corrective action plan (e.g., training and reporting systems) and remeasuring PSC annually to track progress.

A national PSC survey could help benchmark hospitals across Ethiopia -which I am not sure is being done- while international comparisons (e.g., with LMICs/HICs) may identify best practices.

Future qualitative studies could explore why ICU scores lag and how midwives perceive safety differently.

Thank you.

Reviewer #6: General Comments

This manuscript addresses a critical and timely issue—patient safety culture in resource-limited healthcare settings. The authors have gathered empirical insights through a multi-center study in Eastern Ethiopia, utilizing the HSoPSC 2.0 survey instrument. The manuscript is well-written and organized, employing a methodological approach that combines quantitative analysis with qualitative insights. The statistical analyses conducted, including ordinal logistic regression and robust ANOVA techniques, enhance the credibility of the findings.

However, several methodological, analytical, and interpretative elements need clarification or enhancement to maximize their contribution. The revisions suggested below will strengthen transparency, reproducibility, and practical applicability.

Survey Instrument

The selection of the HSoPSC 2.0 tool is appropriate and aligns well with international patient safety research practices. However, the manuscript currently provides limited information regarding its adaptation for use in Ethiopia. The authors are encouraged to give a detailed account of the adaptation process, including any cultural and linguistic adjustments made and the formal validation procedures, such as cognitive testing or pilot implementation, that were undertaken. This information is crucial given the low internal consistency noted in several subscales (Cronbach's alpha <0.6) and will significantly enhance the reproducibility and robustness of the study.

Data Collection Methods

The authors have provided a clear outline of the data collection procedures, which enhances the manuscript's transparency. However, additional details regarding the consent process, particularly how verbal consent was standardized and documented, are needed to bolster the study's validity further. Moreover, clarity on whether survey responses were anonymous or coded would help readers assess potential biases. The training procedures for research assistants also require further clarification to ensure that other researchers can fully replicate the data collection methods.

Data Availability

The manuscript states that data will be available upon reasonable request. However, PLOS ONE strongly encourages or requires public sharing of underlying data to ensure transparency, reproducibility, and open science principles. Depositing de-identified datasets into a publicly accessible repository or providing clear justification if restrictions apply would significantly enhance the manuscript.

Sampling Strategy and Hospital Selection

The manuscript would greatly benefit from additional clarity regarding the participating hospitals' selection criteria. Whether hospitals were chosen for geographic diversity, operational capacity, patient volume, or convenience is unclear. Clarifying the rationale behind hospital selection will help readers better assess the generalizability and validity of the study's comparisons across clinical units and hospitals.

Sociodemographic Characteristics

The authors provide clear and descriptive demographic characteristics of their sample. However, the sampling frame itself needs further explanation. Given that only 686 individuals were approached from a larger potential pool (>1,500 healthcare professionals), additional clarification is required regarding how this specific subset was determined and whether the sample was proportionally representative across roles, departments, or shifts. A more detailed account of the sampling frame will help readers understand potential selection bias and interpret findings.

Dimensions of Patient Safety Culture

The manuscript provides thorough and detailed reporting of composite and item-level scores, consistent with AHRQ recommendations. However, the significant variability within certain composites, particularly Staffing and Work Pace, requires deeper interpretative discussion. The authors should explicitly acknowledge the suboptimal internal reliability of multiple dimensions as a methodological limitation and discuss how it may impact the validity of the conclusions.

Benchmarking

A critical omission in the manuscript is benchmarking against international or regional norms derived from the AHRQ HSoPSC 2.0 database. Without these comparisons, whether the overall score (47% positive response) indicates severe deficiencies or is consistent with broader LMIC or global patterns remains unclear. Benchmarking the results against available international or regional data is crucial as it would significantly enhance the manuscript's contribution by providing essential context for interpreting findings.

Patient Safety Rating and Incident Reporting

The high rate of non-reporting (56%) identified in the manuscript raises essential concerns about reporting culture. The authors are encouraged to provide a more in-depth analysis exploring why underreporting is prevalent. Integrating these findings with qualitative responses and related subscales (such as Response to Error and Communication Openness) would offer valuable insight. The authors should discuss underreporting explicitly in terms of organizational or cultural factors, such as fear of blame or punitive responses, to add interpretative depth and practical implications.

Comparison Across Clinical Units

The methodological rigor in comparing clinical units, particularly the use of Welch's ANOVA and Games-Howell post hoc tests, is commendable. The consistent finding of lower patient safety scores in ICUs is highly significant. To enhance interpretative value, the authors should expand the discussion of these findings by referencing ICU-specific factors such as workload intensity, staff-patient ratios, complexity of care, and institutional resources in Ethiopia. This deeper contextualization will strengthen practical relevance.

Predictors of Safety Ratings

The ordinal regression analysis is clear, rigorous, and appropriate for the study's aims. However, reporting of model fit statistics (e.g., pseudo R²) would enhance methodological transparency and interpretative confidence. Additionally, the significant finding that midwives report lower safety ratings than other professional groups merits deeper exploration. The authors are encouraged to interpret this finding by discussing professional roles, workload distribution, decision-making autonomy, and resource allocation for midwives, highlighting practical intervention opportunities.

Open-Ended Responses

Including open-ended responses enriches the manuscript with qualitative insights, reinforcing key quantitative findings such as resource scarcity and punitive safety culture. However, the manuscript would benefit from more straightforward thematic integration of these qualitative comments into the broader discussion. Explicitly connecting these comments to quantitative findings will help illustrate real-world impacts and improve the manuscript's depth and applicability.

Discussion and Literature Engagement

The authors engage effectively with international literature. However, engagement with prior Ethiopian studies on patient safety using earlier versions of the HSOPSC tool is somewhat limited. The manuscript would significantly benefit from a deeper integration of previous national studies to identify progress or stagnation in patient safety culture, providing valuable local context and continuity in the discussion.

Implications and Conclusion

The implications section is well-developed and offers thoughtful recommendations. However, to fully explore the study's implications, authors are encouraged to prioritize recommendations based on feasibility, urgency, and cost-effectiveness to enhance practical utility. Identifying immediate, actionable interventions suitable for resource-limited settings will increase the manuscript's practical value. Additionally, the conclusion should be sharpened to avoid repetition, emphasizing key findings, unique contributions, and clear, actionable recommendations, particularly concerning ICU-specific strategies and leadership roles.

Reproducibility Note

An overarching concern is the limited methodological detail, especially concerning survey adaptation, hospital selection, and data collection protocols. Expanding these sections to ensure other researchers can fully replicate the study methods is critical for scientific rigor, transparency, and broader applicability of the findings.

Overall Recommendation

This manuscript is valuable to the literature on patient safety culture in resource-constrained settings. With revisions focused on benchmarking, methodological transparency, thematic integration, and deeper contextual interpretation, it can substantially impact practice, policy, and research. The requested revisions are intended to support the authors in strengthening the manuscript's rigor and maximizing its scholarly and practical contributions.

6. PLOS authors have the option to publish the peer review history of their article (what does this mean?). If published, this will include your full peer review and any attached files.

Reviewer #1: **Yes: **Sharifa Alblooshi

Reviewer #2: **Yes: **Jihen Sahli

Reviewer #3: No

Reviewer #4: **Yes: **Collins Atta Poku

Reviewer #5: **Yes: **Ahmed Newera

Reviewer #6: No

---

## [Author Response · Author response to Decision Letter 1]

12 May 2025

Point-by-point response to Editor and Reviewer’s Comments

Editor and reviewers Comments Author responses

Editor’s comment Please address the reviewers' comments, especially those related to benchmarking, methodological robustness and transparency and thematic integration. Thank you for your valuable feedback, which has been instrumental in improving the quality of our manuscript. We have carefully and thoroughly addressed each of the reviewers' comments point-by-point and have incorporated all necessary revisions into the updated manuscript, which has been labelled as 'Manuscript' in the submission system.

Reviewer #1 I would like to thank you for submitting this insightful paper on patient safety culture. This is an incredibly important topic, and your research provides valuable perspectives that can help advance the understanding and improvement of safety protocols in healthcare settings.

The paper is well-written and highlights several crucial aspects of patient safety. I do have a few minor comments, specifically regarding the sample size and some potential gaps in the research that could be explored further.

Your feedback is highly appreciated and has been instrumental in refining our work. We have provided the responses point-by-point as follows;

1. Sample size and sampling technique

Would you please clarify how the sample size was proportionally allocated to each hospital based on the total number of staff and how much was given to each hospital. Thank you for this important question. The total sample size was proportionally allocated across the five participating hospitals based on the total number of healthcare professionals employed at each site using the formula: Sample from hospital A=(Total number of healthcare professionals in Hospital A/total number of healthcare professionals in all hospitals)*Total sample size (686). This proportional allocation ensured that larger hospitals contributed a correspondingly greater share of participants, thereby enhancing the representativeness of the sample. To improve the clarity and transparency on this regard we have included the total number of healthcare professionals at each hospital, and the proportionally allocated sample to each hospital as follows” Accordingly, from Hospital A, out of a total of 612 healthcare professionals, 288 were proportionally allocated. For Hospital B, 186 professionals were available and 88 were allocated; Hospital C had 165 professionals with 77 allocated; Hospital D had 308 with 145 allocated; and Hospital E had 187 with 88 allocated.”(Line 155-160).

2. Conclusion

It is worth stating the gaps this study has covered and any other gaps that should be investigated in the future. We sincerely appreciate your insightful feedback. We have refined the conclusion section and included the contributions of our study (lines 632–642), and the recommendations for future research es including the longitudinal studies to track a change in patient safety culture over time and their impact on patient outcome, in-depth qualitative studies exploring why patient safety culture is lower in intensive care units compared to other units?, psychometric evaluations of the Hospital Survey on Patient Safety Culture 2.0 (HSoPSC 2.0) tool, interventional studies to examine effectiveness of strategies to enhance patient safety culture in resource limited settings along with follow-up studies to assess their scalability and sustainability in such settings. The details of recommendations for future research are highlighted from Line 620-627.

Reviewer #2 The paper provides valuable insights into patient safety culture in resource-limited settings, particularly in Ethiopian public hospitals. The study is well-designed, with a correct methodology responding to the study objectives. However, there are some limitations such as: Your feedback is highly appreciated and has been instrumental in refining our work. We also appreciate your recognition of the importance of addressing these limitations.

• The low Internal Consistency of Subscales: Several subscales in the HSoPSC 2.0 showed poor internal consistency (Cronbach’s α < 0.70), which may have affected the reliability of the results. This limitation is acknowledged by the authors but could impact the validity of the findings. We thank the reviewer for highlighting this important issue. We agree that low internal consistency in several subscales—particularly Staffing and Work Pace (α = .19) and Handoffs and Information Exchange (α = .48)—poses a significant methodological limitation. In response, we have revised the limitations section to explicitly state how this measurement issue may have affected the results. Specifically, we now clarify that these low alpha values may have obscured true effects in both the ANOVA and ordinal logistic regression analyses, potentially resulting in non-significant findings where meaningful relationships may exist. This measurement error increases the risk of Type II errors and may have led to the underestimation of effect sizes. The revised discussion (Lines 554-569) reflects this concern in detail and recommends future scale refinement.

Language Barrier : Although English is the medium of instruction in Ethiopian hospitals, some participants may have faced challenges in understanding the questionnaire, potentially affecting their responses. We agree that language has the potential to influence how questions may have been answered and had included this as a key limitation of our work. We appreciate the reviewer’s recognition of this important issue.

Reviewer #3 Thank you to look over my insight full expertise on this research work entitled “Patient safety culture in resource-limited settings: insights from healthcare professionals through a multi-center survey.” and I got this article is very interesting and thought full and identifying deferent hidden factors that identified Patient safety culture in resource-limited settings.

The research work is very interesting and thought full for scholars and also leading factor for other researchers and also hospital administrators farther more for ministry of health

Thank you for your feedback. We highly appreciate and it has been instrumental in refining our work.

1. How do you measure Cronbach’s α and what is the cut point of Cronbach’s α? The Cronbach’s alpha to assess the reliability of patient safety culture dimensions were calculated by using the using IBM SPSS Statistics for Windows version 30.0 (IBM Corp., Armonk, NY, USA) as outlined in Table 2. Cronbach’s alpha is a coefficient that estimates the proportion of variance in a scale score that is attributable to the true score, rather than to measurement error. It is calculated based on the average inter-item correlations and the number of items in the scale. Interpretation, values of Cronbach’s alpha are generally evaluated as follows:

 ≥ .90 = Excellent

 .80–.89 = Good

 .70–.79 = Acceptable

 .60–.69 = Questionable

< .60 = Poor

While many researchers adopt a cut-off of ≥ .70 as acceptable reliability (Fornell & Larcker, 1981; Hair et al., 2019), available at: https://doi.org/10.1177/002224378101800313 and https://thuvienso.hoasen.edu.vn/handle/123456789/10308 respectively. The developers of the Hospital Survey on Patient Safety Culture (HSoPSC 2.0) consider a Cronbach’s alpha of ≥ .60 to be acceptable for their instrument, given the conceptual breadth and diversity of items within each dimension (AHRQ User Guide). Our evaluation followed this guidance, which is consistent with psychometric practices for broad constructs.

2. What is your base line that you said that the patient safety is present or absent in the hospital? with reference. Thank you for raising this important point. According to the HSoPSC 2.0 user’s guide there is no baseline to categorise patient safety culture is “present” or “absent” in the hospital. However, the patient safety rating can be categorised as developed (>75% positive responses), poorly developed (50–75% positive responses), and underdeveloped (<50% positive responses). Available at: https://doi.org/10.1186/s40886-017-0062-9 and https://www.ahrq.gov/sops/surveys/hospital/index.html. Therefore, we have followed the user’s guide recommendations to categorise the patient safety culture as outlined in the manuscript Line 240-242.

3. Why you were select the linear model for your research? Clearly explain We sincerely appreciate your thoughtful feedback. To clarify, we did not use a linear regression model, as our outcome variable—patient safety rating—was measured on an ordinal Likert scale ranging from “poor” to “excellent.” A linear model would have been inappropriate, as it assumes a continuous dependent variable with equal intervals between values. Instead, we employed ordinal logistic regression (OLR), which is specifically suited for ordinal outcomes because it models the log-odds of being at or below a certain category while preserving the rank order of the responses.

We selected OLR after carefully evaluating its assumptions. We confirmed no multicollinearity among predictors (VIF < 5 and pairwise correlations r < .66), and the proportional odds assumption was supported by the parallel lines test (p > .05). This analytic approach allowed us to answer the research question regarding which factors predict more positive patient safety ratings, in a statistically appropriate and robust manner (Line 249-256).

4. In line 285 your finding “Compared with supervisors and other clinical staff, midwives were significantly less likely to rate patient safety positively,” why this has happened? Would you explain the tool you used (if you can tool). Thank you for your feedback. As noted, midwives were significantly less likely than supervisors and other clinical staff to rate patient safety positively, with an adjusted odds ratio (AOR) indicating a 79.2% lower likelihood (AOR = 0.208, 95% CI: 0.060–0.713, p = 0.013). This finding was based on the ordinal logistic regression (OLR) analysis, interpreted according to the adjusted odds ratio output. We used the HSOPSC 2.0 tool, categorising professional roles as nurses, physicians, midwives, pharmacy and laboratory technicians, supervisors, and other clinical staff. The bivariate OLR analysis identified a significant association (p < 0.05), and this variable was subsequently included in the multivariate OLR model, as presented in Table 4.

5. Do you think that “lack of a patient safety incident reporting system,” is the main challenge is this how do you measure? Thank you for your feedback. The lack of a patient safety incident reporting system was identified as a major challenge to ensuring patient safety in Ethiopian public hospitals. This finding emerged from the open-ended comment section of the HSOPSC 2.0. We thematically categorised the responses, with 17 participants (28%) specifically mentioning the absence of an incident reporting system (Line 360).

6. I do have question on your exclusion criteria do you think that the absent of study participant during study period exclusion criteria? Thank you for your valuable feedback. We agree that, given our use of consecutive sampling, which included all healthcare professionals who were available and consented to participate during the data collection period—the exclusion of individuals on extended leave (e.g., annual, sick, or maternity leave) occurred by default. This is due to the paper-based nature of the survey, which required the physical presence of participants in healthcare facilities. Accordingly, we have removed the corresponding statement from the manuscript stating, “Healthcare professionals on extended leave (e.g., annual, sick, or maternity leave) during the data collection period were excluded”.

7. Do you think is it possible two ethical clearance for one research work? Thank you for your feedback. As the research team included members from two distinct institutions across two countries (Ethiopia and Australia), and the first author was enrolled at an Australian university while conducting the study in Ethiopia, it was necessary to obtain ethical approval from both institutions. There was no conflict between the two ethics approvals, as both adhered to international human research ethics guidelines, ensuring alignment and compliance with ethical standards.

General comment for researchers: I have read your finding, and which is very good work, but I recommended your study population was patients because the key stakeholder due to understand or answer your research title the actors were better to take patients. Thank you for your recommendations. Given the HSoPSC 2.0 tool is designed only for healthcare providers, it was not possible to include the patients as participants in this study. We also recognised the dual importance of healthcare providers, patients (and families) in maintaining patient safety. The recommendation to incorporate the patient perspective in assessing patient safety culture in healthcare facilities for future research is valid. For future work we will consider using standard tool such as Patient Measure of Safety (PMOS) tool to examine the patient perspectives on patient safety. We have also undertaken qualitative interviews with patients and families to explore patient safety concepts and specifically incident reporting in Ethiopian hospitals; this work will be published separately. In this study we have incorporated the recommendations to involve patients and families at policy and practice level to enhance safety and quality in healthcare (Line 443 to 445 & Line 587).

Reviewer #4 Dear sir,

The study examined ‘Patient safety culture in resource-limited settings: insights from healthcare professionals through a multi-centre survey’. This manuscript is well-written and sound, offering a significant understanding of patient safety culture in Ethiopian public hospitals. The study effectively identifies gaps in patient safety culture and provides actionable recommendations for improving it. However, several areas could be enhanced. Overall, with minor improvements, this manuscript has the potential to contribute significantly to the field of patient safety, particularly in resource-limited settings.

We sincerely appreciate your insightful feedback.

Introduction

The transition from HSoPSC 1.0 to 2.0 is mentioned, but the discussion on how HSoPSC 2.0 improves explicitly upon the previous version could be expanded. More concrete examples of its improvements, such as added dimensions or clarified questions, would enhance the reader’s understanding. We have added a statement outlining the improvements made in updating the HSOPSC 1.0 to HSOPSC 2.0 to enhance clarity for the reader. These improvements included wording clarity, a reduction in the number of negatively worded items, a decrease in the number of items from 42 to 32, a reduction in dimensions from 12 to 10, and the incorporation of contemporary perspectives on patient safety (Line 89 to 96).

Again, while the introduction mentions Ethiopia’s under-researched state, a deeper exploration of the Ethiopian healthcare system’s specific challenges could enrich the introduction. For instance, referencing infrastructure deficits, regional disparities, and the socio-political climate in Ethiopia might provide more context for the gap in research. Thank you for your feedback. We agree that including the challenges facing the Ethiopian healthcare system enhances clarity for readers. Accordingly, we have added a statement highlighting “Systemic challenges such as inadequate infrastructure, qualified workforce shortages, weak governance, limited training opportunities, and disparities in awareness further undermine consistent safety standards and contribute to avoidable harm. Ethiopia's healthcare system, in particular, faces significant challenges including limited funding, underdeveloped infrastructure, persistent health inequities, pronounced regional disparities, and recurring socio-political instabilities, placing considerable pressure on delivering equitable, safe, and quality healthcare” (Lines 104–111).

Methods and Materials

1. The use of “consecutive sampling” is mentioned, but the rationale behind this choice could be better explained. For example, it could clarify why thi

---

## [Decision Letter · Decision Letter 1]

29 May 2025

Patient safety culture in resource-limited healthcare settings: A multicentre survey.

PONE-D-25-07741R1

Dear Dr. Fekadu,

We’re pleased to inform you that your manuscript has been judged scientifically suitable for publication and will be formally accepted for publication once it meets all outstanding technical requirements.

Kind regards,

Hossam Elamir, MSc

Academic Editor

PLOS ONE

Additional Editor Comments (optional):

Reviewers' comments:

Reviewer's Responses to Questions

**Comments to the Author**

1. If the authors have adequately addressed your comments raised in a previous round of review and you feel that this manuscript is now acceptable for publication, you may indicate that here to bypass the “Comments to the Author” section, enter your conflict of interest statement in the “Confidential to Editor” section, and submit your "Accept" recommendation.

Reviewer #4: All comments have been addressed

Reviewer #6: All comments have been addressed

2. Is the manuscript technically sound, and do the data support the conclusions?

Reviewer #4: Yes

Reviewer #6: Yes

3. Has the statistical analysis been performed appropriately and rigorously? 

Reviewer #4: Yes

Reviewer #6: Yes

4. Have the authors made all data underlying the findings in their manuscript fully available?

Reviewer #4: Yes

Reviewer #6: No

5. Is the manuscript presented in an intelligible fashion and written in standard English?

Reviewer #4: Yes

Reviewer #6: Yes

6. Review Comments to the Author

Reviewer #4: Well done, congratulations.

The write-up has been copy-edited and the language is clear, correct, and unambigous.

All comments have been addressed

Reviewer #6: Thank you for your thoughtful and comprehensive revision of the manuscript. The updated version demonstrates substantial improvements across several important areas that were highlighted in the initial review. In particular, the enhanced methodological transparency, clearer explanation of the adaptation process for the HSoPSC 2.0 tool, and more detailed justification of the sampling strategy have significantly strengthened the scientific rigor and replicability of the study.

Benchmarking against international and regional studies was a major concern in the previous round, and the integration of comparative data has added valuable context to your findings. Your expanded discussion around ICU-specific challenges and the professional differences in safety culture perception—especially among midwives—adds interpretative depth and relevance.

The qualitative integration of open-ended responses is now more thematically connected to the quantitative results, providing a richer understanding of the safety culture landscape in resource-limited hospital settings. Additionally, your refined practical recommendations offer more concrete guidance for decision-makers, with appropriate consideration of resource constraints.

While the data availability statement could be further aligned with PLOS ONE’s open data policy by providing unrestricted access or public repository links, you have acknowledged the issue transparently.

Overall, this revised manuscript represents a meaningful and well-articulated contribution to the patient safety literature, particularly within LMIC contexts. Thank you for your careful revisions and commitment to improving the clarity and rigor of your work.

7. PLOS authors have the option to publish the peer review history of their article (what does this mean?). If published, this will include your full peer review and any attached files.

Reviewer #4: **Yes: **Poku, Collins Atta

Reviewer #6: No

---

## [Editor Report · Acceptance letter]

PONE-D-25-07741R1

PLOS ONE

Dear Dr. Fekadu,

I'm pleased to inform you that your manuscript has been deemed suitable for publication in PLOS ONE. Congratulations! Your manuscript is now being handed over to our production team.

Kind regards,

on behalf of

Dr. Hossam Elamir

Academic Editor

PLOS ONE